# Regular Exercise in *Drosophila* Prevents Age-Related Cardiac Dysfunction Caused by High Fat and Heart-Specific Knockdown of *skd*

**DOI:** 10.3390/ijms24021216

**Published:** 2023-01-07

**Authors:** Yurou Cao, Shiyi He, Meng Ding, Wenzhi Gu, Tongquan Wang, Shihu Zhang, Jiadong Feng, Qiufang Li, Lan Zheng

**Affiliations:** Key Laboratory of Physical Fitness and Exercise Rehabilitation of Hunan Province, College of Physical Education, Hunan Normal University, Changsha 410012, China

**Keywords:** *skd*, exercise, HFD, heart function, lipid metabolism, *Drosophila*, metabolic syndrome

## Abstract

*Skuld (skd)* is a subunit of the Mediator complex subunit complex. In the heart, *skd* controls systemic obesity, is involved in systemic energy metabolism, and is closely linked to cardiac function and aging. However, it is unclear whether the effect of cardiac *skd* on cardiac energy metabolism affects cardiac function. We found that cardiac-specific knockdown of *skd* showed impaired cardiac function, metabolic impairment, and premature aging. *Drosophila* was subjected to an exercise and high-fat diet (HFD) intervention to explore the effects of exercise on cardiac *skd* expression and cardiac function in HFD *Drosophila*. We found that Hand-Gal_4_>*skd* RNAi (KC) *Drosophila* had impaired cardiac function, metabolic impairment, and premature aging. Regular exercise significantly improved cardiac function and metabolism and delayed aging in HFD KC *Drosophila*. Thus, our study found that the effect of *skd* on cardiac energy metabolism in the heart affected cardiac function. Exercise may counteract age-related cardiac dysfunction and metabolic disturbances caused by HFD and heart-specific knockdown of *skd*. *Skd* may be a potential therapeutic target for heart disease.

## 1. Introduction

The global obesity epidemic continues to grow relentlessly and now affects over two billion people [1,2]. Obesity increases the risk of cardiovascular disease [3], raises the chances of metabolic disorders [4], and increases blood lipids [5]. There are various factors that contribute to obesity, such as high-fat diet (HFD) intake, aging, and a sedentary lifestyle [6]. HFD can lead to lipid accumulation, conduction block, and severe structural lesions in the heart [7]. Studies have reported that the heart can fine-tune metabolism and gene expression patterns to adapt to dietary changes and external stimuli to maximize energy efficiency [8,9]. Cardiac energy metabolism affects systemic energy metabolism and homeostasis in vivo [10,11,12]. The heart under normal conditions relies primarily on the oxidation of fatty acids delivered by the circulation for energy [13,14]. *Skuld* (*skd*) is primarily expressed in the heart, skeletal muscle, and brain in human tissues [15]. *Skd* is homologous to the human Mediator complex subunit 13 (MED13) [16]; the gene sequence position is 3L:20992814..21027434.

Cardiac-specific knockdown of *skd* in *Drosophila* increases fat accumulation and induces obesity. Cardiac *skd* is involved in whole-body energy metabolism [10] and is associated with obesity and other diseases related to energy metabolism [11]. Studies have shown that cardiac-specific knockout of *skd* increases susceptibility to accumulation of triglycerides (TG) and obesity in HFD mice [17]. In particular, there are sex differences in *skd* expression, and *skd* is a protective factor in the hearts of healthy female rats [18]. In addition, cardiac *skd* regulates other transcription factors (TFs) involved in the regulation of lipid biosynthesis and metabolism, such as *srebp* [19] and Eip75B [20]. *Skd* may be a possible therapeutic target for metabolic and nonmetabolic heart disease. However, it remains unclear whether the effect of cardiac *skd* on cardiac energy metabolism affects cardiac function. Further studies are needed to determine whether cardiac *skd* is associated with other cardiovascular diseases and whether it regulates cardiac function. In addition, cardiac brummer (bmm) has been identified as a key antagonist of HFD-induced lipotoxic cardiomyopathy in *Drosophila* [7,21]. The metabolic dysregulation associated with obesity caused by poor lifestyle habits is similar to that observed in normal aging, and there is substantial evidence that obesity has the potential to accelerate aging. Adipose tissue storage, distribution, and function change with age. Moreover, adipose tissue is associated with aging and aging-related diseases, such as heart disease [22,23,24] and metabolic dysfunction [25].

Exercise training reduces the incidence of obesity and heart disease [26,27,28,29], and slows aging [30]. Our previous studies have shown that regular exercise improves cardiac function [31,32], counteracts myocardial lipotropic damage induced by HFDs, and leads to a benign reversal of cardiac lipid metabolism [33,34,35]. A growing number of studies have shown that regular exercise at different times of life also delays cardiac age-related phenotypes [36,37,38]. Thus, the potential mechanism of exercise resistance to HFD-induced age-related premature heart failure remains to be further determined.

*Drosophila* is a well-known model organism for the study of basic metabolic diseases such as obesity and diabetes [39]. *Drosophila* fed HFD for short periods of time show lipid accumulation, structural changes, and premature failure in the myocardium. This study explored the effect of the *skd* gene intervention in the heart on cardiac function and whether cardiac *skd* is further damaging to cardiac energy metabolism and cardiac function under HFD conditions.

We wanted to understand whether exercise can modulate *skd* expression in the heart to improve whole-body energy metabolism and cardiac function in *Drosophila* and prevent premature cardiac failure caused by HFD and cardiac-specific knockdown of *skd*. Hence, we sought to investigate the relationship between exercise, HFD, *skd*, and cardiac aging.

## 2. Results

### 2.1. High-Fat Diet Promotes Aging-Related Cardiac Dysfunction and Systemic Lipid Accumulation

Similar to mammals, *Drosophila* developed an age-related decline in cardiac function with reduced heart rate and increased fibrillation particularly pronounced under stress conditions [40]. It is unclear whether *skd* gene expression changes in the heart with aging and whether *skd* in the heart affects cardiac function. In addition, these longitudinal muscles are closely associated with the heart tube, and have been proposed to assist in circulation of the hemolymph through the heart tube [41]. In this experiment, to determine the adverse effects of aging on the cytoarchitecture of myocytes, a 12-day (12D) and 36-day (36D) old *Drosophila* cardiac tubules were assayed for filamentous actin (F-actin) structures in myogenic fibers using Phalloidin. As shown in Figure 1A, senescent myocardial fibers are less dense and disorganized, the regularity of their myogenic fibers is reduced, and the heart tube is thinner. We examined cardiac function using M-mode. Compared to Hand-Gal_4_>w^1118^-12D-NFD-C (12-C), Hand-Gal_4_>w^1118^-36D-NFD-C (36-C) shows significant increases in heart period (HP) (Appendix A), arrhythmia index (AI) (Figure 1B,C), systolic intervals (SI) (Appendix A), and diastolic intervals (DI) (Appendix A), and a significant decrease in heart rate (HR) (Appendix A). Thus, it was shown that aging leads to cardiac dysfunction in *Drosophila*. qPCR further confirmed the relationship between the expression level of *skd* in the heart tube and aging, and the results show that the relative mRNA expression of *skd* in the heart tube of 36-C *Drosophila* was also significantly lower than that of 12-C *Drosophila* (Figure 1D).

However, in mammals, HFD causes systemic lipid overload, which accelerates the aging of their organs and affects cardiac function, leading to severe heart failure [33,42]. *Drosophila* fed an HFD in which 30% coconut oil is added to the normal medium for 5 days of HFD [7,43], is the widely used model for inducing their obesity. We performed HFD interventions in young and old *Drosophila* by staining for ghost pen cyclic peptides to observe the structure of F-actin in myogenic fibers after HFD. The myocardial fibers of HFD-fed *Drosophila melanogaster* show a similar phenotype to that of senescence-reduced density, disorganized arrangement, reduced regularity of their myogenic fibers, and changes in the morphology of the heart tube (Figure 1A). Analysis of the cardiac function revealed that the cardiac function was impaired. HR (Appendix A) and AI (Figure 1B,C) increased significantly, and HP (Appendix A), DI (Appendix A), and SI (Appendix A) decreased significantly in Hand-Gal_4_>w^1118^-12D-HFD-C (12-HC) *Drosophila*. Hand-Gal_4_>w^1118^-36D-HFD-C (36-HC) *Drosophila* shows clear signs of irregular heartbeat (Figure 1C), a significant increase in HP (Appendix A), a significant decrease in HR (Appendix A), and a significant increase in DI (Appendix A). In addition, the AI (Figure 1B,C) and SI (Appendix A) show a nonsignificant increase. The 12-HC AI and 36-C AI values were essentially the same, suggesting that HFD accelerates *Drosophila* aging. However, does HFD affect *skd* gene expression in the heart? The relationship between the cardiac *skd* and HFD was examined using qPCR, which shows that the relative mRNA level expression of *skd* in the heart tube was significantly reduced in both 12D and 36D *Drosophila* after HFD intervention (Figure 1D).

TG are the major form of lipid storage in mammals and *Drosophila*, and elevated TG causes obesity and metabolic syndrome. In rodent models of obesity and diabetes and in human subjects, myocardial steatosis is associated with functional and structural changes in the heart [44]. Our results indicate that the 36-C *Drosophila* shows increased adipose area and TG levels compared to the 12-C *Drosophila*. In the 12-HC and 36-HC *Drosophila*, Oil red O (ORO) test results show a significant increase in abdominal fat (Figure 1E) and whole-body TG (Figure 1F). Thus, after 5 days of feeding HFD to 12D or 36D *Drosophila*, cardiac function is impaired and whole-body fat accumulation is increased.

Both senescent and HFD *Drosophila* show lipid accumulation, so is metabolism within the heart tube altered? We went on to examine changes in a number of metabolic genes in the *Drosophila* heart tube to elucidate the mechanisms underlying senescence, HFD, and obesity levels. Compared with the 12-C *Drosophila*, in the 36-C *Drosophila*, the relative mRNA expression levels of bmm (Figure 1G) decreased significantly, and the relative mRNA expression levels of *srebp* (Figure 1H) and Eip75B (Figure 1I) in the heart tube increased, but it was not significant. Under HFD conditions, the relative mRNA expression levels of *srebp* (Figure 1H) and Eip75B (Figure 1I) increased significantly in the heart tube of *Drosophila* at 12-HC, while the relative mRNA expression levels of bmm (Figure 1G) decreased significantly. At 36-HC, the relative mRNA expression levels of srebp (Figure 1H) and Eip75B (Figure 1I) increased significantly in the heart tube of *Drosophila*, while the relative mRNA expression levels of bmm (Figure 1G) increased, but it was not significant.

In conclusion, HFD affects cardiac function, disrupts cardiac structure, leads to obesity, and accelerates aging in *Drosophila*. Both HFD and aging lead to an age-related increase in arrhythmias, which may be due to the downregulation of the expression of cardiac *skd* and metabolic disorders. However, the effect of the cardiac *skd* gene on the heart is not well understood.

### 2.2. Cardiac Skd-Specific RNAi and/or HFD Promotes Age-Related Cardiac Insufficiency and Increased Systemic Lipids

Systemic TG increased significantly after knockdown of *Drosophila skd* using cardiac-specific Tin-Gal_4_ [17]. However, the relevance of the relationship between the cardiac *skd* gene and cardiac function is unclear. In this experiment, Hand-Gal_4_>*skd* RNAi (KC) *Drosophila* were constructed using the UAS/Hand-gal_4_ system to test whether cardiac-specific knockdown of *skd* impairs their cardiac function. The results showed that *skd* gene expression levels were lower in the hearts of Hand-Gal_4_>*skd* RNAi-12D-NFD-C (12-KC) *Drosophila* than in the control 12-C *Drosophila*, indicating that the KC *Drosophila* strain was successfully constructed (Figure 2A). However, genetic manipulation of the cardiac myocytes may have a significant non-cell-autonomous effect on the ventral muscle layer containing longitudinal myogenic fibers. To determine whether the specific knockdown of *skd* mRNA levels and fat accumulation in KC *Drosophila* hearts adversely affect the cytoarchitecture of cardiac myocytes, the heart tubes of KC were stained with ghost pen cyclic peptide. The results show that the cardiac structure of the 12-KC *Drosophila* was disrupted, with reduced density and disorganized arrangement of the cardiac fibers and gaps in certain locations (Figure 2B). This implies that the cardiac-specific knockdown of *skd* affects myocardial development in *Drosophila*. Further analysis of cardiac function shows a significant increase in AI (Figure 2C,D), a significant decrease in HP(Appendix A) and SI (Appendix A), and no significant change in DI(Appendix A) and HR(Appendix A) in the 12-KC *Drosophila* compared to the 12-C *Drosophila*.

In *Drosophila*, HFD downregulates *skd* gene expression in the heart, promotes total body fat gain, and accelerates aging. Heart-specific knockdown of *skd* leads to an increase in total body lipids, so does it lead to premature aging? To explore the relationship between 12C-KC and HFD and heart and senescence, first, 12-HC was compared with 12-KC. The heart morphology of the 12-KC *Drosophila* was similar to the heart tube morphology of the 12-HC *Drosophila* (Figure 1A and Figure 2B). Furthermore, by analyzing the heart function, it was found that HR (Appendix A), HP (Appendix A), AI (Figure 2C,D), DI (Appendix A), and SI (Appendix A) of 12-KC were not significantly different from those of 12-HC. 12-KC and 12-HC show the same phenotype. Cardiac-specific knockdown of *skd* appeared to have similar effects to HFD.

Further HFD intervention on 12-KC resulted in a significant decrease in the density of Hand-Gal_4_>*skd* RNAi-12D-HFD-C (12-KHC) *Drosophila* cardiac myocardial fibers, whose arrangement appeared very disorganized and even fiber loss in some places and thinning of the heart tube diameter (Figure 2B). Further analysis of the cardiac function shows that HR reduced significantly in 12-KHC (Appendix A), and AI (Figure 2C,D), HP (Appendix A), SI (Appendix A), and DI (Appendix A) increased significantly. We determined the relationship between the expression level of *skd* and HFD in the heart using qPCR. The expression of *skd* relative mRNA in the 12 KHC heart reduced significantly (Figure 2A). We further demonstrate that HFD and heart-specific knockdown of *skd* after *Drosophila* appear to have aging-like characteristics.

To explore fat accumulation in KC *Drosophila*, we examined TG levels and ORO. The results show that TG levels increased significantly in the 12-KC *Drosophila* compared with the 12-C *Drosophila* (Figure 2F) and ORO staining was stronger (Figure 2E). We further tested whether cardiac-specific knockdown of *skd* promotes fat accumulation in *Drosophila* under HFD conditions. The 12-KHC *Drosophila* shows a significant increase in abdominal fat (Figure 2E) and a significant increase in whole-body TG levels (Figure 2F).

We further investigated the expression levels of metabolic genes and *skd*-regulated transcription factors in the heart tube of Hand-Gal_4_>*skd* RNAi *Drosophila* to explore the mechanisms by which *skd* promotes age-related cardiac insufficiency and increases systemic lipids. The relative mRNA expression levels of *srebp* (Figure 2G) and bmm (Figure 2H) were significantly lower, and that of Eip75B (Figure 2I) was significantly higher in 12-KC, compared with 12-C. Under HFD conditions, the relative mRNA expression levels of 12-KHC *srebp* were significantly higher (Figure 2G), and the relative mRNA expression levels of Eip75B (Figure 2I) and bmm (Figure 2H) decreased significantly.

These results suggest that cardiac *skd*-specific RNAi and/or HFD promote age-related cardiac insufficiency, increased systemic lipids, and promote aging in the *Drosophila* heart. Under HFD conditions, aging was accelerated in KC *Drosophila*, further impairing cardiac function. The mechanism may be related to altered levels of cardiac *srebp*, Eip75B, and bmm mRNA expression.

In addition, a two-way ANOVA was performed with *skd* RNAi (C vs. KC) and HFD (HFD and NFD) as factors, as shown in Table 1. The 12 D Hand-Gal_4_>*skd* RNAi *Drosophila* show a significant interaction between physical exercise and HFD in terms of HP (*p* = 0.000) and HR (*p* = 0.000) and DI (*p* = 0.000) and SI (*p* = 0.008) significant interactions, and no significant interactions were detected for AI and TG.

### 2.3. Regular Exercise Resists HFD-Induced Decline in Cardiac Skd Expression, Age-Related Cardiac Abnormalities, and Increased Systemic Lipid Accumulation

Exercise training improves cardiac function, metabolic disorders, and fat loss in *Drosophila* and regulates metabolic gene expression in the heart [32]. To investigate the effect of regular exercise at different ages on cardiac function and metabolism, we exercised young and old *Drosophila* and found that exercise had an effect in both young and old age. First, the results from ghost pen cyclin show a more regular arrangement and increased density of cardiac myogenic fibers in the Hand-Gal_4_>w^1118^-12D-NFD-E (12-E) and 36-E *Drosophila* (Figure 3A). Cardiac function reduced significantly in AI (Figure 3B,C) and HR (Appendix A) in the 12-E group compared with those in the 12-C group. There were significant increases in DI (Appendix A) and HP (Appendix A) and no significant changes in SI (Appendix A). Compared with 36-C, SI (Appendix A) and AI (Figure 3B,D) were significantly lower and highly significantly different, HR (Appendix A) increased significantly, and HP (Appendix A) and DI (Appendix A) did not change significantly in Hand-Gal_4_>w^1118^-36D-NFD-E (36-E). Further testing using qPCR shows that the *skd* gene was significantly upregulated in the heart tube in both 12-E and 36-E *Drosophila* compared with the sedentary group (Figure 3E,F).

In addition, the effect of exercise under different dietary conditions was investigated using phalloidin staining. The results show a more regular arrangement and increased density of cardiac myogenic fibers in Hand-Gal_4_>w^1118^-12D-HFD-E (12-HE) and Hand-Gal_4_>w^1118^-36D-HFD-E (36-HE) *Drosophila* (Figure 3A). Further analysis of the cardiac function revealed a significant decrease in AI (Figure 3B,D) and HR (Appendix A) and a significant increase in DI (Appendix A), SI (Appendix A), and HP (Appendix A) in 12-HE compared with 12-HC. Compared with 36-HC *Drosophila*, AI (Figure 3B,D), HP (Appendix A), and DI (Appendix A) decreased significantly, HR (Appendix A) increased significantly, and SI (Appendix A) was not significantly different in 36-HE. Further detection using qPCR shows that the *skd* gene was significantly upregulated in the heart tube in both NFD and HFD exercise groups, compared with the sedentary group, in 12-E, 12-HE, 36-E, and 36-HE *Drosophila* (Figure 3E,F).

To further observe the improving effect of exercise on lipids in *Drosophila*, the 12-E and 36-E *Drosophila* were tested for ORO and TG levels. The results show that in the NFD situation, 12-E had reduced abdominal fat (Figure 3G) and significantly increased whole-body TG levels (Figure 3H). In the 36-E *Drosophila*, their whole-body TG levels reduced significantly (Figure 3I), and their abdominal ORO shows that fat deposition became less (Figure 3G). In the case of HFD, the 12-HE and 36-HE *Drosophila* the abdominal ORO shows that abdominal fat deposition became less (Figure 3G) and the whole-body TG levels decreased significantly (Figure 3H,I). Moreover, To further our explanation of the mechanisms of the exercise effect, the mRNA expression levels of the relevant metabolic genes in the *Drosophila* heart tube were examined. The results show that the relative mRNA expression levels of *srebp* (Figure 3J,K) and Eip75B (Figure 3L,M) reduced significantly following the exercise intervention in W^1118^>Hand-Gal_4_
*Drosophila* under either 12D or 36D and NFD or HFD conditions. Compared with 12-C and 12-HC, the expression levels were significantly higher for both 12-E and 12-HE bmm (Figure 3N); 36-E bmm was significantly higher than 36-C, and 36-HE bmm was significantly lower than 36-HC (Figure 3O).

Therefore, regular exercise resisted the HFD-induced decrease in cardiac *skd* expression, age-related cardiac function abnormalities, and increased systemic lipid accumulation, suggesting that exercise can resist an increase in AI due to aging and HFD, and can improve cardiac function in *Drosophila*. The mechanisms may be that exercise improves the mRNA levels of *skd* and related metabolic genes in the *Drosophila* heart.

### 2.4. Regular Exercise Improves Age-Related Cardiac Dysfunction and Systemic Lipid Increase Caused by Cardiac Skd-Specific RNAi

To investigate whether regular exercise can counteract the age-related cardiac dysfunction and systemic lipid increase induced by cardiac *skd*-specific RNAi. First, we explored the ameliorative effects of regular exercise on cardiac function in the KC *Drosophila*. The morphology of the heart tube was restored in the Hand-Gal_4_>*skd*^RNAi^-12D-NFD–E (12-KE) *Drosophila*, the arrangement of myogenic fibers became regular, and their number increased (Figure 4A), and the gap of the fibers reduced. Further analysis of heart function shows that exercise significantly reduced HR (Appendix A) and AI (Figure 4B,C) and significantly increased HP (Appendix A), DI (Appendix A), and SI (Appendix A). The relative expression levels of *skd* mRNA in *Drosophila* heart tubes increased significantly (Figure 4D).

We further explored the effect of regular exercise on improving cardiac function in HFD-fed KC *Drosophila*. The morphology of the heart tube improved, and the arrangement of myogenic fibers became regular and increased in number in the Hand-Gal_4_>*skd*^RNAi^-12D-NFD–C (12-KHE) *Drosophila* after regular exercise (Figure 4A). From the results of the cardiac function analysis, 12-KHE shows significantly reduced AI (Figure 4B,C), HP (Appendix A), DI (Appendix A), and SI (Appendix A) and significantly increased *skd* content in the heart tube (Figure 4D).

In addition, with cardiac-specific knockdown of the *skd* gene, *Drosophila* developed an obese phenotype. This phenotype improved with exercise. The 12-KE and 12-KHE *Drosophila* show a significant reduction in abdominal fat (Figure 4E) and a significant reduction in TG levels (Figure 4F).

To further determine the mechanisms by which exercise ameliorates cardiac dysfunction in KC and the obesity phenotype in *Drosophila*, bmm, *srebp*, and Eip75B mRNA levels were further examined in the heart tube of the KC *Drosophila*. Under NFD conditions, the relative mRNA expression levels of 12-KE, *srebp* (Figure 4G), and Eip75B (Figure 4H) decreased significantly and the relative mRNA expression levels of bmm increased significantly (Figure 4I). Under HFD conditions, the relative mRNA expression levels of 12-KHE, *srebp* (Figure 4G), and Eip75B (Figure 4H) reduced significantly and bmm relative mRNA expression levels increased significantly (Figure 4I).

The results of this experiment show that exercise can counteract cardiac dysfunction caused by cardiac-specific knockdown of *skd*, mainly improving rhythm-related indicators in *Drosophila*. Exercise also improved the systematic obesity and cardiometabolic status of the KC *Drosophila* and resisted premature aging. Thus, exercise training improves cardiac dysfunction and obesity due to the cardiac-specific knockdown of *skd*, and the mechanisms may be related to the upregulation of cardiac *skd* gene expression and metabolic genes.

## 3. Discussion

HFD can cause obesity and metabolic disorders. Obesity and metabolic disorders cause cardiac dysfunction and a pathophysiological phenotype of myocardial response to triacylglycerol dysregulation. In contrast, *skd* in the heart controls metabolic homeostasis, and the specific knockdown of *skd* in the heart of *Drosophila* increases fat accumulation and induces obesity [10,17]. It has recently been reported that *skd* may be a possible therapeutic target for metabolic and nonmetabolic heart disease. However, it is unknown whether the effect of *skd* on cardiac energy metabolism affects cardiac function [11]. Furthermore, cardiac brummer (bmm) has been identified as a key antagonist of HFD-induced lipotropic cardiomyopathy in *Drosophila* [7,21]. Cardiac *skd* regulates the transcription factors involved in the regulation of lipid biosynthesis and metabolism, such as SREBP and Eip75B (Eip75B is a PPARγ homolog in *Drosophila*). In the present study conducted separately in 12D and 36D *Drosophila*, we discussed whether the effect of cardiac *skd* on cardiac energy metabolism affects cardiac function, and whether regular exercise modulates cardiac *skd* against age-related cardiac dysfunction and metabolic impairment induced by HFD in *Drosophila*.

We have revealed the effects of cardiac *skd* on cardiac function. The results show that the specific knockdown of cardiac *skd* causes general obesity and abnormal cardio metabolism in *Drosophila* and affects the cardiac function, mainly by affecting rhythm and aging-like characteristics. Under HFD conditions in Hand-Gal_4_>*skd* RNAi flies, the specific knockdown of cardiac *skd* further impairs cardiac function in response to HFD. Cardiac *skd* mRNA levels decreased with advancing age. Regular exercise under different intervention conditions and at different ages resisted both *skd* RNAi and HFD-induced cardiac impairment and metabolic disturbances. Changes in the mRNA expression levels of *srebp*, Eip75B, and bmm were observed under different intervention conditions and at different ages. These results suggest a previously unknown relationship between *skd* and cardiac function, and suggest that exercise may improve the cardiac dysfunction caused by HFD and aging.

*Drosophila* is a very popular model organism in the study of metabolism and aging. It has a tubular heart with a conserved mechanism of heart tube development [40,45,46]. Approximately 77% of human disease genes can be found in *Drosophila*, including genes associated with cardiovascular diseases [40,45,47]. It has a well-established and rich set of molecular and genetic tools [48], technological tools [49], and is short-lived [38,50]. Exercise has been shown to improve cardiac function and high-fat-induced metabolic disorders in *Drosophila*. Therefore, we used *Drosophila* as a model organism to explore the relationship between *skd*, HFD, aging, and exercise.

One study showed obesity phenotypes in rodents fed HFD [51]. Diet-induced obesity usually induces many secondary diseases, including heart disease [52]. Epidemiologists have linked HFD to heart diseases [53]. For example, *skd* has been shown to play an important role in regulating energy metabolism in the heart of *Drosophila* and mice [8,10,17]. Chronic caloric excess leads to increased delivery of fat-derived fatty acids and cytokines to the heart and skeletal muscle, which may increase the risk of organ lipotoxicity [54,55,56]. Short-term feeding of *Drosophila* with HFDs affects their cardiac function, with lipid accumulation and changes in myocardial structure [7]. HFDs appear to have direct lipotoxic effects on the myocardium. Fibrillation is also seen in *Drosophila*, and it is similar [57] and a common persistent arrhythmia [58]. Saturated fatty acids can poison normal cells. Both lard and coconut oil are rich in saturated fatty acids. Saturated fatty acids induce apoptosis of the ventricular cardiomyocytes, activation of stress-related protein kinases, and oxidative stress of proteins [59]. The main component of olive oil is monounsaturated fatty acids. It has been shown that extra virgin olive oil diets significantly improved glycemia, insulinemia, glucose tolerance, insulin sensitivity, and insulin degradation. At a later stage, we can consider the use of different oils for the configuration of the high-fat medium [60]. In humans and *Drosophila*, an important marker of aging is the ectopic deposition of fat, leading to impaired multi-organ function and metabolic changes [31,50,61,62]. In turn, HFD causes obesity, which accelerates aging and is associated with the induction of age-related diseases such as cardiovascular disease [63]. Many studies have shown that metabolism changes with age [50].

The results of this study show that HFD in *Drosophila* increased TG levels and severe degeneration and disruption of myogenic fibers during the different life cycles. In addition, aging caused a significant decrease in the expression of the *skd* gene in the heart, a decrease in cardiac function, structural damage to the heart, blurring of myocardial fibers, and metabolic disturbances, in line with previous experiments. In both obese and diabetic rodent models and human subjects, myocardial steatosis is associated with functional and structural changes in the heart [44,64]. In addition, cardiac aging in mammals is always accompanied by disorganized and reduced myocardial fiber arrangement [65].

*Skd* is involved in systemic energy metabolism and has been linked to obesity, diabetes, and other diseases related to energy metabolism [11]. Whole-body energy metabolism is regulated in the hearts of *Drosophila* and mouse [17]. Cardiac overexpression of *skd* in transgenic (α-myosin heavy chain (αMHC)-MED13-TG or MED13-cTG) mice inhibits the binding of RNA polymerase II (Pol II) to MED, thereby inhibiting the transcription and expression of target genes involved in cardiac energy metabolism, improving insulin sensitivity and increasing energy expenditure, thereby preventing HFD-induced obesity [66]. In *Drosophila*, *skd* has been identified as a negative regulator of lipid accumulation [18].

In rodents, cardiac *skd* deficiency in the context of hypothyroidism exacerbates cardiac dysfunction [67]. Cardiac *skd* expression is primarily involved in energy homeostasis. However, it is unclear whether the effect of *skd* on cardiac energy metabolism affects cardiac function. Further studies are needed to determine whether cardiac *skd* is associated with other cardiovascular diseases and whether it regulates cardiac function.

We have demonstrated that cardiac-specific knockdown of *skd* affects cardiac function, mainly affecting cardiac rhythm problems, through a significant increase in AI and changes in cardiac structure, with the appearance of depression. Under NFD conditions, 12-KC showed a generalized obesity phenotype, increased systemic triglyceride levels, increased cardiac AI in *Drosophila*, and cardiac dysfunction in *Drosophila*. Moreover, the phenotype of 12-KC was similar to that of 12-HC and 36-C. Under HFD conditions, *skd* expression levels were reduced in the heart tube of 12-KHC *Drosophila*, and mRNA expression levels of srebp, Eip75B, and bmm were altered in the heart tube.

Our results show that HFD, aging, and cardiac-specific knockdown *skd* increase cardiac AI in *Drosophila*. The degree of fibrosis and fat accumulation increases in the aging *Drosophila* heart. Senescence is a disruption of intracellular Ca^2+^ [68] regulation driven by changes in the kinetics of L-type Ca^2+^ channel currents [69], transient outward K channel currents, and reduced activity of the SERCA Ca^2+^ pump [70]. The combination of these mechanisms ultimately leads to a high degree of arrhythmia.

Then, A study reported that a HFD activated NADPH oxidase 2 (NOX2) in the heart, which increased oxidative stress and led to abnormal calcium handling [71]. HFD can reduce Kv1.5, which generates repolarizing currents. Abnormal repolarization appears to play a role in the pathophysiology of arrhythmias after HFD [72]. In addition, high lipids exacerbate lipid accumulation and alter the cardiac structure and environment, and we showed that HFD altered *Drosophila* heart structure, HR, HP, DI, and SI. HFD promotes obesity and activates key signaling pathways including the renin–angiotensin–aldosterone system, TGF-β, connective tissue growth factor, and endothelin-1 activation can lead to increased interstitial collagen deposition and cardiac tubular fibrosis, which may disrupt atrial conduction, leading to AI [73,74].

Cardiac-specific knockdown *skd Drosophila* show a similar phenotype of hyperlipidemia and senescence with increased AI, probably also due to the above.

Exercise is an economical, nontoxic, and effective pill to improve heart function, alleviate and prevent obesity, improve metabolic levels, and delay premature aging. Some studies have reported that exercise can reduce excess body fat accumulation and obesity [75]. Exercise is considered an effective way to delay cardiac aging in the form of physiological stress that it induces. In aging mammals, increasing evidence confirms that moderate exercise training reduces abnormal cardiac remodeling, left-ventricular dilatation, myocardial fibrosis, mitochondrial dysfunction, and cardiac dysfunction, and improves cardiac function and quality of life [18,66]. Previous studies in our laboratory have confirmed that exercise improves cardiac function in *Drosophila* [32,33,35,38,76].

Our results suggest that exercise improves HFD, aging, and cardiac-specific knockdown *skd Drosophila* heart structure and reduces AI. The reduction of cardiac AI by exercise may be due to improvements in cardiometabolic risk factors, reduced sympathetic drive, and favorable changes in cardiac structure and function [73,77].

Exercise improves cardiac function in high-fat *Drosophila*, but it is unclear whether regular exercise improves age-related cardiac dysfunction and systemic lipid increase caused by cardiac *skd*-specific RNAi. To demonstrate this, we fed W^1118^>Hand-Gal_4_ and Hand-Gal_4_>*skd* RNAi flies HFD for 5 days, followed by regular exercise. The results showed that exercise increased *skd* expression in the heart tube, reduced AI, improved cardiac rhythm disturbances and cardiac myogenic fiber reconstruction in *Drosophila* in both the 12D and 36D groups, and substantially rescued myocardial fibers. Whole-body TG levels and ORO staining intensity were reduced. Curiously, whole-body TG increased in 12-E and 12-KE *Drosophila*; this may be because 12D *Drosophila* is in a growth phase, and the body is stressed to store energy to incubate offspring, causing the body to store fat. It is also possible that exercise raises the number of eggs and increases the size of the ovaries in *Drosophila*. This is because lipid accumulation is also important in reproduction and is necessary for ovarian fat accumulation in female *Drosophila* [76].

HFD downregulates *skd* expression in the heart tube of *Drosophila* 12D and 36D, with stronger ORO intensity and increased TG levels. Aging decreases *skd* expression in the heart tube, and increases ORO intensity, TG levels, and body weight. Exercise was also found to increase *skd* expression levels and reduce heart mass in mice [78]. The results showed that exercise training upregulated *skd* expression levels in the heart in both schizont and *Drosophila*.

The strategy of regulating substrate utilization to improve oxidative metabolism is rapidly becoming a popular therapy. Thus, can exercise improve metabolic genes? Exercise improves the expression levels of *skd*, srebp, Eip75B, and bmm in the heart at different ages and under different dietary conditions. In contrast, lipolysis, a major component of lipid metabolism, is the breakdown of TG to free fatty acids. *Drosophila* brummer (bmm) is homologous to the human adipose triglyceride lipase [79]. Previous studies have shown that bmm overexpression effectively prevents HFD-induced lipid accumulation and cardiac dysfunction [7]. *Srebp* sterol regulatory element binding proteins (SREBPs) are the major transcription factors that regulate cholesterol, fatty acid, and triglyceride biosynthesis [80], Inhibition of SREBP decreased cholesterol and fatty acid biosynthesis [81] and studies have reported that SREBPs partially mediate the deleterious effects of HFD on the heart. *Drosophila* Eip75B is homologous to the human and mammalian peroxisome value-added agent-activated receptor PPAR-γ. Studies have shown that cardiac-specific overexpression of Eip75B induces lipotoxicity in the heart [82]. Reduced activity of Eip75B, increased FFA oxidation, and reduced abiogenesis resist HFD-induced obesity [83]. Our results show that bmm mRNA levels are decreased in the hearts of HFD, aging, and heart-specific knockdown *skd Drosophila*, and increased in the heart tube after exercise. In the gene normal expression group, cardiac tubular srebp and Eip75B mRNA levels were elevated after HFD intervention and aging. In the NFD, 12-KC *Drosophila* heart tube Eip75B mRNA levels were similarly elevated, as were 12-KHC *Drosophila* heart srebp mRNA levels. Under NFD and HFD conditions, *srebp* and Eip75B mRNA levels in the heart tube were decreased in *Drosophila* from the genetically normal and cardiac deliberate knockdown groups after regular exercise. Unexpectedly, bmm mRNA levels were elevated in 12-KHC *Drosophila*. Possibly, the elevated triglyceride levels themselves in cardiac-specific knockdown *Drosophila* further exacerbate lipid accumulation after hyperlipidemia, possibly activating their own protective mechanisms and upregulating bmm expression levels. As with bmm, srebp and Eip75B mRNA levels mRNA expression levels were decreased in the heart tube of 12-KC *Drosophila*. The reason may be that the lipid content of *Drosophila* increased after *Drosophila* heart-specific knockdown of *skd*, but the young *Drosophila*’s own regulatory mechanism was not completely disrupted, and self-protection occurred. Decreased bmm mRNA expression levels in the heart tube of *Drosophila* 36-KE. The reason for this may be that triglycerides are a medium lipid storage form, and in older *Drosophila*, fat stores become less and less, while exercise promotes organism health and resists aging [84], keeping *Drosophila* heart tube bmm mRNA levels in a normal state.

Thus, exercise improved various physical indicators, and the mechanisms could be that exercise improved the mRNA expression levels of *skd*, srebp, Eip75B, and bmm in the heart tube. Based on our findings, we conclude that whenever we consume an HFD, it can impair our cardiac function and metabolic homeostasis. Appropriate exercise is effective and can improve cardiac function and metabolic status whether started young or old. These results suggest that specific knockdown of *skd* in the *Drosophila* heart tube causes general obesity, cardiometabolic disorders, cardiac dysfunction, and structural disruption of the heart, with loss of myogenic fibers and disorganization. Regular exercise increases *skd* expression in the heart, counteracts age-related cardiac and metabolic disorders induced by cardiac-specific knockdown of *skd* in HFD-fed *Drosophila*, restores the structure of the *Drosophila* heart tube and rescues myogenic fibers, and improves obesity caused by diet and aging.

However, in this experiment, a 5-day exercise intervention was performed on 12D and 36D *Drosophila*, with the possibility of a later exercise intervention of a different duration and different exercise lengths to further explore the mechanisms of exercise.

It is worthwhile to further explore the mechanisms by which regular exercise ameliorates age-related cardiac dysfunction caused by cardiac *skd*-specific knockdown. It may be related to the reported involvement of *skd* in adipose tissue conversion. Further proof at the protein side at a later stage. In addition, the *Drosophila* model also has limitations in human cardiovascular disease. Morphologically, the heart of *Drosophila* is a simple linear tube, while that of human beings is circular. Functionally, the *Drosophila* heart pumps hemolymph in the open circulatory system, while the human heart is part of the closed circulatory system [40]. The subtle physiological defects observed in the human cardiovascular system cannot be completely reproduced in *Drosophila* [85]. In *Drosophila*, it is still impossible to observe the reduction in the number of myocardial cells and left-ventricular hypertrophy in human heart aging.

A characteristic of human cardiac aging is a reduction in the number of cardiomyocytes, as well as a moderate left-ventricular hypertrophy [86]. However, it is difficult to determine the equivalent of “heart hypertrophy” in the aged hearts of flies.

In conclusion, HFD impairs cardiac function and promotes metabolic disorders. Cardiac-specific knockdown of *skd* impairs cardiac function, disrupts lipid metabolism, and accelerates aging in *Drosophila*. In contrast, exercise is the remedy, and regular exercise can counteract the age-related cardiac dysfunction caused by high lipids and cardiac *skd*-specific knockdown.

## 4. Materials and Methods

### 4.1. Fly Stocks and Groups

Two strains of *Drosophila*, W^1118^ (stock number: 3605; genotype: W^1118^) and hand-Gal_4_ (stock number: 48396; W ^[1118]^; P{y ^[+t7.7]^ w ^[+mC]^ = GMR88D05-GAL_4_} attP2), were purchased from Bloomington *Drosophila* Stock Center. *Skd-UAS-RNAi* (stock number: v330726; genotype: P {VSH330726} attP40) was purchased from the Vienna *Drosophila* RNAi Center. The control *Drosophila* and cardiomyocyte *skd* knockdown *Drosophila* were generated by crossing male Hand-Gal_4_ with female W^1118^ or UAS-*skd* RNAi, and all crosses were collected within 12 h from F_1_-featured females for the experiment.

There are three variables in this experiment. For ease of writing, high-fat diet is denoted by HFD, normal-fat diet is denoted by NFD, regular exercise is denoted by E, 12 days old is denoted by 12D, and 36 days old is denoted by 36D. This experiment has the following groupings: Hand-Gal_4_>w^1118^-12D-NFD-C (12-C), Hand-Gal_4_>w^1118^-12D-NFD-E (12-E), Hand-Gal_4_>w^1118^-12D-HFD-C (12-HC), Hand-Gal_4_>w^1118^-12D-HFD-E (12-HE), Hand-Gal_4_>w^1118^-36D-NFD-C (36-C), Hand-Gal_4_>w^1118^-36D-NFD-E (36-E), Hand-Gal_4_>w^1118^-36D-HFD-C (36-HC), Hand-Gal_4_>w^1118^-36D-HFD-E (36-HE), Hand-Gal_4_>*skd*RNAi-12D-NFD-C (12-KC), Hand-Gal_4_>s*kd*RNAi-12D-NFD-E (12-KE), Hand-Gal_4_>*skd*RNAi-12D-HFD-C (12-KHC), and Hand-Gal_4_>*skd*RNAi-12D-HFD-E (12 -KHE).

During the experiment, NFD *Drosophila* were placed in a 25 ± 1 °C incubator with 50% humidity and a 12 h light/dark cycle; HFD-fed *Drosophila* were placed in a 22 °C incubator to prevent coconut oil from melting and sticking to the flies.

### 4.2. Diet Preparation

The medium was prepared as Ryan T Birse [7,35]. The normal medium consists of soybean flour, corn flour, yeast flour, and sucrose. The high-fat medium was prepared by adding 30% coconut oil to the normal medium and mixing it well. All the diets were changed once every 2 days. 12D *Drosophila* started HFD on 2D and ended on 6 D; 36D *Drosophila* started HFD on 26D and ended on 30D.

### 4.3. Motion Devices and Protocols

We invented a locomotor device to induce continuous upward movement of the fruit flies by exploiting their natural negative grounding behavior. A glass jar containing 20 fruit flies was fixed horizontally to a steel tube, which is rotated along its horizontal axis by an electric motor, with a gear that regulates its shaft speed. The glass tube rotates with the tube, causing the fruit flies in the glass tube to make a climbing motion. Most of the fruit flies responded by climbing throughout the movement, while the few that could not climb actively walked along the inner wall of the glass tube. The distance between the glass jar and the stopper was adjusted before exercise to ensure that each glass tube was 8 cm away for exercise. In addition, the environment during exercise was consistent with the incubator environment. All exercise groups in this experiment had an exercise intervention of 5 D for 2.5 h each [87]. The 12-day-old exercise group started exercise on 7 D and ended on 11 D, with pickup on 12 D. The 36 D exercise group started exercise on 31 D and ended on 35 D, with pickup on 36 D [36].

### 4.4. Real-Time Quantitative PCR (qPCR)

Thirty heart tube tissue samples were collected from each group and homogenized, and total RNA was extracted using Trizol (Invitrogen, CA, USA) reagent in lysate according to the manufacturer’s protocol. cDNA was generated using Superscript III reverse transcriptase (Invitrogen, CA) and used as a template for quantitative real-time PCR. Thermal cycling and fluorescence monitoring was performed in an ABI7300 Real-time PCR Instrument (Applied Biosystems, USA): (30 s at 95 °C, 5 s at 95 °C, and 30 s at 60 °C) × 40. Real-time PCR was performed using SYBR green (TaKaRa) normalized with rp49. The relative abundance of the genes tested was calculated using the 2^−ΔΔCt^ method [88]. The primers used for the expression analysis were *skd* (F: TCCCATAGCCGAGAAGATCCTTGAG; R: CTTATGACCACCCGACACCACTTC); *srebp* (F: GCAGTTCCTTCGTTTTCTTTTC; R: GGCTTCCATTTCCAGTCAGTT); bmm (F: ACTCACATTTCGCTTACCC; R: GAGAATCCGGGTATGAAGCA); Eip75B (F: AACTGCACCACCACTTGACA; R: TTCTTCTCGTTGCCCGACTC); Rp49 (F: CTAAGCTGTCGCACAAATGG; R: AACTTCTTGAATCCGGTGGG).

### 4.5. Analysis of Cardiac Function

Steps for preparation of semi-exposed *Drosophila* hearts: First, an artificial hemolymph solution was configured and placed under 26 °C conditions to pump oxygen for 30 min. Then, the fruit flies were anesthetized with carbon dioxide for 2–3 min and fixed on Petri dishes coated with medical petroleum jelly. Then, the head was removed with special scissors and forceps at the microscope, and the artificial hemolymph fluid filled with oxygen was added. Finally, the ventral thorax and ventral abdominal corpuscles were cut open under the microscope with special scissors and forceps to clip out their internal organs. The surrounding fat was aspirated with a capillary needle to expose the heart, and oxygen was pumped continuously for 30 min [89].

The heartbeat of the *Drosophila* was recorded using an EM-CCD high-speed camera (video 124 fps for 30 s), and the ECG data were recorded using HC Image software. Analysis of *Drosophila* heart function based on previous research methods by Martin Fink [90]. Semi-automated optical heartbeat analysis was used to quantify the heart rate (HR), heart period (HP), diastolic intervals (DI), systolic intervals (SI), and arrhythmia index (AI). Each sample group was 25 ± 5.

### 4.6. Oil Red O (ORO)

ORO is a fat-soluble dye that specifically colors neutral fats such as TG in the tissue. Sample preparation: The *Drosophila* was dissected in ADH and the head and tail were removed leaving the back plate. Sample fixation: The PBS was discarded, and 4% paraformaldehyde was used to fix the sample for 25 min. First wash: The 4% paraformaldehyde was discarded, and the sample was washed three times/10 min with PBS. Staining: A drop of Oil red O reagent (Oil Red O solution3:2 for staining solution and distilled water) was placed on the sample and incubated for 30 min on a shaker. Second wash: The dye was discarded, and the sample was washed with PBS three times/10 min. Sample fixation: A slide was prepared, and then the sample was transferred onto the slide and covered with a coverslip. Photography: A Leica stereo microscope (Leica; Wetzlar; Germany) was used to capture the images.

### 4.7. Phalloidin

Semi-intact *Drosophila* hearts were prepared as described previously [89]. The ADH was quickly replaced with a relaxation buffer (containing 10 mM EGTA ADH), and 4% paraformaldehyde was used to fix the sample for 25 min. Washing: The 4% paraformaldehyde was discarded, and the sample was washed three times/10 min with PBS. Staining: Drops of ghost pen cyclopeptide dye were placed on the sample and incubated for 30 min on a shaker protected from light. Third washing: The dye was discarded, and the sample was washed with PBS three times/10 min. Sample fixation: A slide was prepared, and then the sample was transferred to the prepared slide and covered with a coverslip. Finally, a Leica stereo microscope (Leica; Wetzlar; Germany) was used to capture the images.

### 4.8. Triglyceride Determination

The microplate method was used to measure whole-body TG levels in *Drosophila*. The test was performed using the Shanghai Enzyme-linked Biotechnology Co., Ltd. Triglyceride Content Assay Kit (ml076637, China). Take 15 fruit flies of appropriate age, divide them into 3 groups, and put them into mortar, 1 mL of anhydrous ethanol was added, homogenized on ice, centrifuged at 12,000 rpm for 10 min at 4 °C, and the supernatant was taken for measurement. This was used strictly according to the manufacturer’s instructions [43].

### 4.9. Statistical Analysis

The independent-samples *t* tests were used to assess the differences between 12-C Drosophila and 36-C, 12-C Drosophila and 12-HC Drosophila, 36-C Drosophila and 36-HC, 12-C and 12-KC, 12-KC and 12-KHC, and the sedentary group of Drosophila and the exercise group. Two-factor ANOVA was used, followed by LSD tests among 12-C, 12-HC, 12-KC, and 12-KHC. The GraphPad Prism and Statistical Package for Social Sciences (SPSS) version 2.0 were used for statistical analysis. The significance was set at *p* < 0.05. All the data are presented as means ± SEM [31].

### 4.10. Institutional Review Board Statement

The animal study protocol was approved by the Biomedical Research Ethics Committee of Hunan Normal University (Ethics Section 2022 No. 450).

## 5. Conclusions

1. Cardiac-specific knockdown of *skd* impairs cardiac function, disrupts lipid metabolism in *Drosophila*.

2. Regular exercise can resist age-related cardiac dysfunction caused by high lipids and cardiac *skd*-specific knockdown.

## Figures and Tables

**Figure 1 ijms-24-01216-f001:**
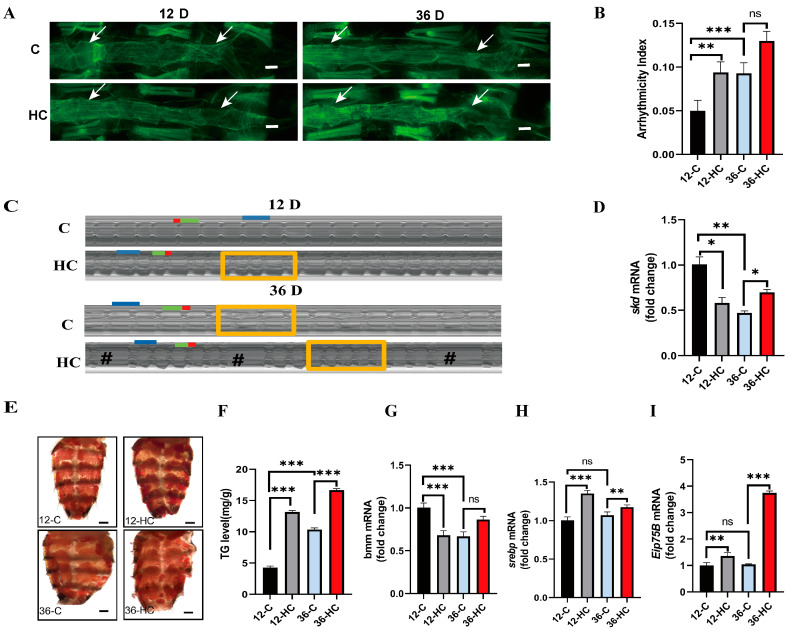
Effects of HFD and aging on cardiac function and systemic TG metabolism in *Drosophila*. (**A**) Cardiac F-actin staining in W^1118^>Hand-Gal_4_
*Drosophila*. Note: scale bar = 100 μm, The two white arrows point to the heart of the fruit fly. (**B**) Cardiac function assays included AI in 12-C, 12-HC, 36-C, and 36-HC groups, N = 25 ± 5. (**C**) M-mode ECG. Note: Cardiac cycle—each group of M-mode ECGs was intercepted for 10 s, N = 25 ± 5. HP: horizontal blue line. DI: horizontal green line. SI: horizontal red line, Rectangle for fibrillation, # for cardiac arrest. (**D**) Cardiac *skd* expression. (**E**) *Drosophila* Abdominal ORO, scale bar = 250 μm. (**F**) Systemic TG of W^1118^>Hand-Gal_4_
*Drosophila*. (**G**–**I**) The cardiac srebp, bmm, and Eip75B mRNA expression levels of 12D W^1118^>Hand-Gal_4_
*Drosophila*. Independent-samples *t* tests were used to assess differences between 12-C *Drosophila* and 36-C *Drosophila*; 12-C *Drosophila* and 12-HC *Drosophila*; 36-C *Drosophila* and 36-HC *Drosophila*. The data represent the mean, and the error bars represent SEM. ns *p* > 0.05; * *p* < 0.05; ** *p* < 0.01; *** *p* < 0.001.

**Figure 2 ijms-24-01216-f002:**
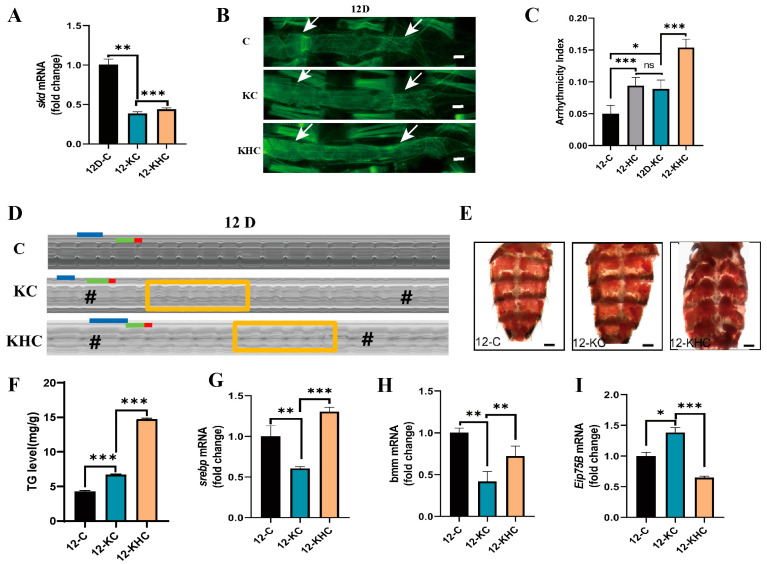
Cardiac function and systemic TG in 12-KC *Drosophila* under NFD and HFD conditions. (**A**) Cardiac-specific knockdown of *skd* in *Drosophila* strain validation and cardiac *skd* expression under 12-KC HFD conditions. (**B**) Cardiac F-actin staining in 12-KC *Drosophila*. Note: scale bar = 100 μm, The two white arrows point to the heart of the fruit fly. (**C**) 12-KC cardiac function under NFD and HFD conditions in *Drosophila* AI. (**D**) M-mode ECG, cardiac cycle—each group of M-mode ECGs was intercepted for 10 s, N = 25 ± 5. HP: horizontal blue line. DI: horizontal green line. SI: horizontal red line. Rectangle for fibrillation. # for cardiac arrest. (**E**) *Drosophila* Abdominal ORO. scale bar = 250 μm, N = 5. (**F**) *Drosophila* Whole-body TG levels. (**G**–**I**) The cardiac *srebp*, bmm, and Eip75B mRNA expression levels on 12D KC *Drosophila*. Two-factor ANOVA was used, followed by LSD tests for 12-C, 12-HC, 12-KC, and 12-KHC. Independent-samples *t* test for assessment of 12-C and 12-KC, 12-KC and 12-KHC TG, SREBP, bmm, and Eip75B levels. The data represent the mean, and the error bars represent SEM. ns *p* > 0.05; * *p* < 0.05; ** *p* < 0.01; *** *p* < 0.001.

**Figure 3 ijms-24-01216-f003:**
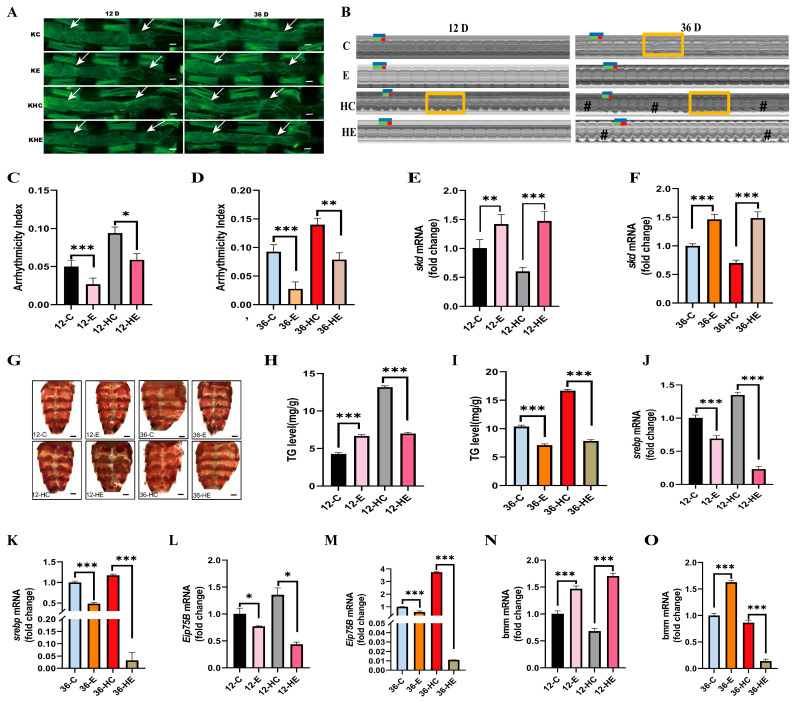
Effects of regular exercise on cardiac function and general obesity of *Drosophila* with HFD. (**A**) Cardiac F-actin staining. Note: scale bar = 100 μm, The two white arrows point to the heart of the fruit fly. N = 5. (**B**) Effects of regular exercise on 12D and 36D NFD and HFD W^1118^>Hand-Gal_4_
*Drosophila* M-mode ECG. Note: Cardiac cycle—each group of M-mode ECGs was intercepted for 10 s, N = 25 ± 5. HP: horizontal blue line. DI: horizontal green line. SI: horizontal red line. Rectangle for fibrillation. # for cardiac arrest. (**C**) Regular exercise affects AI in cardiac function in 12D NFD and HFD *Drosophila*. N = 25 ± 5. (**D**) Regular exercise affects AI in cardiac function in 36D NFD and HFD *Drosophila*. N = 25 ± 5. (**E**) Effect of regular exercise on 12D NFD and HFD W^1118^>Hand-Gal_4_
*Drosophila* heart *skd* mRNA expression levels. (**F**) Effect of regular exercise on 36D NFD and HFD W^1118^>Hand-Gal_4_
*Drosophila* heart *skd* mRNA expression levels. (**G**) Effects of regular exercise on abdominal ORO staining in 12D, 36D NFD, and HFD W^1118^>Hand-Gal_4_.scale bar = 250 μm, N = 5. (**H**) Effects of regular exercise on systemic TG in 12D, NFD and HFD W^1118^>Hand-Gal_4_
*Drosophila*. (**I**) Effects of regular exercise on systemic TG in 36D, NFD and HFD W^1118^>Hand-Gal_4_
*Drosophila*. (**J**) Expression levels of 12D *Drosophila* heart tube *srebp* mRNA in after exercise intervention. (**K**) Expression levels of 36D *Drosophila* heart tube *srebp* mRNA in after exercise intervention. (**L**) Expression levels of 12D *Drosophila* heart tube *Eip75B* mRNA in after exercise intervention. (**M**) Expression levels of 36D *Drosophila* heart tube *Eip75B* mRNA in after exercise intervention. (**N**) Expression levels of 12D *Drosophila* heart tube bmm mRNA in after exercise intervention. (**O**) Expression levels of 36D *Drosophila* heart tube bmm mRNA in after exercise intervention. The independent-samples *t* test was used to assess the difference between the sedentary group of *Drosophila* and the exercise group. The data represent the mean, and the error bars represent SEM. * *p* < 0.05; ** *p* < 0.01; *** *p* < 0.001.

**Figure 4 ijms-24-01216-f004:**
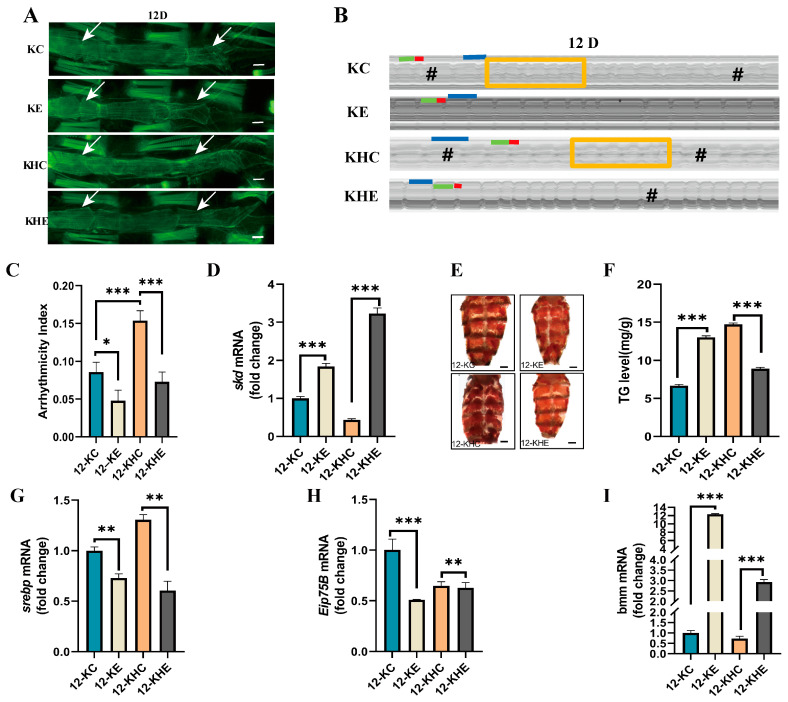
Effects of regular exercise on cardiac function and systemic obesity in Hand-Gal_4_>*skd* RNAi HFD *Drosophila*. (**A**) Cardiac F-actin staining in *Drosophila*. Note: scale bar = 100 μm, The two white arrows point to the heart of the fruit fly. N = 5. (**B**) Effects of regular exercise on cardiac function in 12D NFD and HFD Hand-Gal_4_>*skd* RNAi *Drosophila* M-mode ECG. Note: Cardiac cycle—each group of M-mode ECGs was intercepted for 10 s, N = 25 ± 5. HP: horizontal blue line. DI: horizontal green line. SI: horizontal red line. Rectangle for fibrillation. # for cardiac arrest. (**C**) Effects of regular exercise on AI in cardiac function in 12D Hand-Gal_4_>*skd* RNAi NFD and HFD *Drosophila*. N = 25 ± 5. (**D**) Effects of regular exercise on cardiac *skd* mRNA expression levels in 12D NFD and HFD Hand-Gal_4_>*skd* RNAi *Drosophila*. (**E**) Abdominal ORO staining. scale bar = 250 μm, N = 5. (**F**) *Drosophila* whole-body TG levels. (**G**–**I**) Cardiac tube *srebp*, bmm, and Eip75B mRNA expression levels after exercise intervention in 12D NFD and HFD Hand-Gal_4_>*skd* RNAi *Drosophila*. The independent-samples *t* test was used to assess the difference between the sedentary group of *Drosophila* and the exercise group. * *p* < 0.05; ** *p* < 0.01; *** *p* < 0.001.

**Table 1 ijms-24-01216-t001:** Two-way analysis of variance results for interaction effect of 12D *Drosophila* physical *skd* × HFD.

Dependent Variable	Type III Sum of Squares	df	Mean Square	F	Sig.
Heart rate	7.183	1	7.183	38.747	0.000
Heart period	0.736	1	0.736	56.233	0.000
Arrhythmicity index	0.004	1	0.004	0.856	0.357
Diastolic intervals	0.607	1	0.607	55.378	0.000
Systolic interval	0.006	1	0.006	7.252	0.008
TG level	0.528	1	0.528	4.425	0.069

## Data Availability

Not applicable.

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
