# Peer review of "Regular Exercise in Drosophila Prevents Age-Related Cardiac Dysfunction Caused by High Fat and Heart-Specific Knockdown of skd"

_ijms, 2023, doi:10.3390/ijms24021216_

Round 1
Reviewer 1 Report (Previous Reviewer 2)
accept
Author Response
We sincerely thank the reviewer for thoroughly examining our manuscript. Thanks for your positive feedback on this study. I believe that with your suggestions, the quality of this study has been improved.Thank you.

Reviewer 2 Report (New Reviewer)
This is a revised version MSS entitled ' Regular exercise in Drosophila prevents age-related cardiac dysfunction caused by high fat and heart-specific knockdown of skd' b u Cao Y et al., This is a good piece of work but poorly presented.
Response to previous comments is not addressed properly.
Figure 1 panel A and H are not convincing. remove phalloidin written in green , it should be clearly indicated in legend to figs.
What is the scale bar here for figures is missing or not visible.
Figure 1,Panel H difficult to follow
Same is true about Figure 2 panel A and B and Figure 3 panel A and H
On page 15 line 590 'The methods of previous studies were used' what is the meaning of this is not clear. There are many other places through out this MSS need some correction.
Abbreviation should be in alphabetical order and use abbreviation wisely.
Also, remember every thing is not abbreviation this should be used wisely.
Legends to figures are not well presented.
Author Response
We sincerely thank the reviewer for thoroughly examining our manuscript and providing very helpful comments to guide our revision. We have tried our best to revise the manuscript according to your kind and constructive comments and suggestions. The responses to the comments are given below.
Point 1: Figure 1 panel A and H are not convincing. remove phalloidin written in green , it should be clearly indicated in legend to figs. What is the scale bar here for figures is missing or not visible.Figure 1,Panel H difficult to follow.
Same is true about Figure 2 panel A and B and Figure 3 panel A and H
Response 1: Thanks for your suggestion. Figure 1 panel A we removed phalloidin written in green , which is clearly indicated in legend to figs.The two white arrows point to the heart of the fruit fly. We have bold the scale of Figure 1 panel A. Make the scale clear.
Figure 1,Panel H is M-mode ECG. Note: Cardiac cycle—each group of M-mode ECGs was intercepted for 10 s. HP: horizontal blue line. DI: horizontal green line. SI: horizontal red line, Rectangle for fibrillation, # for cardiac arrest. And we rearranged the M-mode ECGs, and enlarged the picture. At the same time,we thickened thehorizontal blue line, horizontal green line,horizontal red line, rectangle and #.
Figure 2 panel A and B, Figure 3 panel A and H and Figure 4 panel A and B are corrected according to the above method.
Thanks again for your constructive suggestions.
Point 2: On page 15 line 590 'The methods of previous studies were used' what is the meaning of this is not clear. There are many other places through out this MSS need some correction.
Response 2: Thanks for your suggestion.We further described the experimental method.
On page 15 line 572-574 ,”5.2. Diet preparation. The medium was prepared as Ryan T Birse [7, 35]. Normal medium consists of soybean flour, corn flour, yeast flour, sucrose.”
On page 16 line 588-591,”All exercise groups in this experiment had an exercise intervention of 5 D for 2.5 h each[77]. The 12-day-old exercise group started exercise on 7 D and ended on 11 D, with pickup on 12 D. The 36 D exercise group started exercise on 31 D and ended on 35 D, with pickup on 36 D[36].”
On page 16 line 593-594,”Thirty heart tube tissue samples were collected from each group and homogenized, and total RNA was extracted using Trizol (Invitrogen, CA, USA).”
On page 16 line 599,” Real-time PCR was performed using SYBR green (TaKaRa)”
On page 16 line 608-610,”Steps for preparation of semi-exposed Drosophila hearts: First, an artificial hemolymph solution was configured and placed under 26℃conditions to pump oxygen for 30 min. Then, ...”
On page 16 line 615-616,” The surrounding fat was aspirated with a capillary needle to expose the heart, and oxygen was pumped continuously for 30 min[79].”
On page 16 line 619-620,” Analysis of Drosophila heart function based on previous research methods by Martin Fink [80].”
On page 16 line 629-630,” A drop of Oil red O reagent (Oil Red O solution3:2 for staining solution and distilled water) “
On page 17 line 633-634,” A Leica stereo microscope (Leica; Wetzlar; Germany) was used to capture the images.”
On page 17 line 644-645,”a Leica stereo microscope (Leica; Wetzlar; Germany) was used to capture the images.”
On page 17 line 649-650,”Take 15 fruit flies of appropriate age, divide them into 3 groups, and put them into mortar,”
On page 17 line 661,”All the data are presented as means ± SEM[31].”
Thanks again for your constructive suggestions.
Point 3:Abbreviation should be in alphabetical order and use abbreviation wisely.
Also, remember every thing is not abbreviation this should be used wisely.
Response 3: Thanks for your suggestion. We have arranged the abbreviations alphabetically and used them correctly.
On page 18 line 678-689,”Abbreviation section:
AI arrhythmia index
Bmm brummer
DI diastolic intervals
F-actin filamentous actin
HP heart period
HR heart rate
HFD high-fat diet
ORO Oil red O
skd Skuld
SI systolic intervals
TG triglycerides”
Thanks again for your insightful comments.
Point 4: Legends to figures are not well presented.
Response 4: Thanks for your helpful suggestion. We have made a lot of changes to all the pictures in the text as required. The legend below the picture has also been modified.
On page 4 line 152-158,”(A) Cardiac F-actin staining in W1118>Hand-Gal4 Drosophila. Note: The two white arrows point to the heart of the fruit fly. (B-F) Cardiac function assays included HR, HP, AI, DI, SI in 12-C, 12-HC, 36-C and 36-HC group, N = 25 ± 5. (G) Cardiac skd expression. (H) M-mode ECG. Note: Cardiac cycle—each group of M-mode ECGs was intercepted for 10 s, N = 25 ± 5. HP: horizontal blue line. DI: horizontal green line. SI: horizontal red line, Rectangle for fibrillation, # for cardiac arrest. (I) Drosophila Abdominal ORO. (J) Systemic TG of W1118>Hand-Gal4 Drosophila. (K–M) The cardiac srebp, bmm and Eip75B mRNA expression levels of 12D W1118>Hand-Gal4 Drosophila.”
On page 7 line 235-238,”(A) Cardiac F-actin staining in 12-KC Drosophila. Note: The two white arrows point to the heart of the fruit fly. (B) M-mode ECG, cardiac cycle—each group of M-mode ECGs was intercepted for 10 s, N = 25 ± 5. HP: horizontal blue line. DI: horizontal green line. SI: horizontal red line. Rectangle for fibrillation. # for cardiac arrest. (C) Cardiac function in 12KC Drosophila. ‘
On page 7 line 241-243,”(F) Drosophila Abdominal ORO. (G–K)12-KC cardiac function under NFD and HFD conditions in Drosophila HR, HP, AI, DI, and SI. (L) Cardiac function indices, HR, HP, AI, DI, SI in 12-KHC and 36-C Drosophila. (M) Drosophila Whole-body TG levels.”
On page 10 line 304-307,”Figure 3. Effects of regular exercise on cardiac function and general obesity of Drosophila with HFD. (A) Cardiac F-actin staining. Note: The two white arrows point to the heart of the fruit fly . N=5. (B–F) Regular exercise affects HR, HP, AI, DI, and SI in cardiac function in 12D NFD and HFD Drosophila. N=25±5.”
On page 10 line 312-317,“ N=5. (I) Effects of regular exercise on systemic TG in 12D, NFD and HFD W1118>Hand-Gal4 Drosophila . (J) Effects of regular exercise on 12D and 36D NFD and HFD W1118>Hand-Gal4 Drosophila M-mode ECG. Note: Cardiac cycle—each group of M-mode ECGs was intercepted for 10 s, N = 25 ± 5. HP: horizontal blue line. DI: horizontal green line. SI: horizontal red line. Rectangle for fibrillation. # for cardiac arrest. ( K) Effects of regular exercise on systemic TG in 36D, NFD and HFD W1118>Hand-Gal4 Drosophila .”
On page 11-12 line 365-369,”Note: The two white arrows point to the heart of the fruit fly. N=5. (B) Effects of regular exercise on cardiac function in 12D NFD and HFD Hand -Gal4 >skd RNAi Drosophila M-mode ECG. Note: Cardiac cycle—each group of M-mode ECGs was intercepted for 10 s, N = 25 ± 5. HP: horizontal blue line. DI: horizontal green line. SI: horizontal red line. Rectangle for fibrillation. # for cardiac arrest.”
感谢您的有用建议。
我相信,在您的建议下,这项研究的质量得到了提高。我们要感谢审稿人,希望修改后的稿子适合发表。再次非常感谢您的意见和建议。

This manuscript is a resubmission of an earlier submission. The following is a list of the peer review reports and author responses from that submission.
Round 1
Reviewer 1 Report
The authors should be commended on their work, both the magnitude and the written portion. Taken together, this is a pretty compelling story.
In this context, authors should make the next modifications; this would improve the quality of their manuscript.
1) No ethics committee authorization number; please put it in methods section.
2) Where is the abbreviation section?
3) It is also surprising that the authors have not had any limitations in their study. Please provide
4) I was struck by the low resolution of each of the figures. Please, could it be corrected?
5) Please, authors should provide a description of the statistical analysis they have used
6) The authors base their conclusions on analysis of mRNA levels, and we know that this is not sufficient since the expression of mRNA to proteins can be altered by different factors. In other words, the fact that we have a punctual increase in mRNA does not mean that we also have an increase in protein.
Minor comments.
7) Although more of a minor concern, the manuscript should be carefully reviewed for small grammatical errors (especially in the discussion).
Author Response
Response to Reviewer 1 Comments
We sincerely thank the reviewer for thoroughly examining our manuscript and providing very helpful comments to guide our revision. We have tried our best to revise the manuscript according to your kind and constructive comments and suggestions. The responses to the comments are given below.
Point 1: No ethics committee authorization number; please put it in methods section.
Response 1: Thanks for your suggestion. We have added the ethics committee authorization number on Lines 641–643(On page 15).
Thanks again for your constructive suggestions.
Point 2: Where is the abbreviation section?
Response 2: Thanks for the question. We have added it on Lines 656–687 (On pages 15-16).
Point 3: It is also surprising that the authors have not had any limitations in their study. Please provide
Response 3: Thanks for your suggestion. Much work has been done in our study, but there are certainly some shortcomings. In the discussion section, we have reported them on Lines 514–517 (on page 12). “However, in this experiment, a 5-day exercise intervention was performed on 12D and 36D Drosophila, with the possibility of a later exercise intervention of a different duration and different exercise lengths to further explore the mechanisms of exercise.”
In addition, it has been reported in the literature that skd in the heart regulates systemic lipid changes and is involved in the transformation of adipose tissue, suggesting that skd is involved in crosstalk in various organs. These aspects were not addressed in this experiment, and we will investigate them at a later stage. We have added new limitations to the results section of the discussion. On pages 12-13, Lines 529–532. “It is worthwhile to further explore the mechanisms by which regular exercise ameliorates age-related cardiac dysfunction caused by cardiac skd-specific knockdown. It may be related to the reported involvement of skd in adipose tissue conversion. Further proof at the protein side at a later stage.”
Thanks again for your insightful comments.
Point 4: I was struck by the low resolution of each of the figures. Please, could it be corrected?
Response 4: Thanks for your helpful suggestion. We have corrected it.
Point 5: Please, authors should provide a description of the statistical analysis they have used
Response 5: Thanks for your helpful suggestion. We previously had instructions in the notes below each image. Now, we have a separate note on Lines 633–640 (on page 15).
Point 6: The authors base their conclusions on analysis of mRNA levels, and we know that this is not sufficient since the expression of mRNA to proteins can be altered by different factors. In other words, the fact that we have a punctual increase in mRNA does not mean that we also have an increase in protein.
Response 6: Thank you very much for your suggestion. Currently, our results are based on the mRNA levels, which can describe the initial purpose of this experiment. It has been shown that exercise regulates skd 1 expression; the expression levels of bmm 2, srebp 3, and Eip75B 4 also changed under high-fat diet conditions; and the trend of change was consistent with the results of this experiment. In addition, we may have difficulties in performing the experimental supplementation this time due to the relatively tight schedule.And we have purchased several commercial antibodies and found that the specificity is not good and there may be species differences. At a later stage, we plan to prepare the antibody ourselves. However, we will continue the discussion at the protein level in a follow-up study.
Thanks again for your constructive suggestions.
References:
- Fernandes, T.; Barretti, D. L.; Phillips, M. I.; Menezes Oliveira, E., Exercise training prevents obesity-associated disorders: Role of miRNA-208a and MED13. Mol Cell Endocrinol 2018, 476, 148-154.
- Gronke, S.; Mildner, A.; Fellert, S.; Tennagels, N.; Petry, S.; Muller, G.; Jackle, H.; Kuhnlein, R. P., Brummer lipase is an evolutionary conserved fat storage regulator in Drosophila. Cell Metab 2005, 1 (5), 323-30.
- Xiao, J.; Xiong, Y.; Yang, L. T.; Wang, J. Q.; Zhou, Z. M.; Dong, L. W.; Shi, X. J.; Zhao, X.; Luo, J.; Song, B. L., POST1/C12ORF49 regulates the SREBP pathway by promoting site-1 protease maturation. Protein Cell 2021, 12 (4), 279-296.
- Khan, D.; Ara, T.; Ravi, V.; Rajagopal, R.; Tandon, H.; Parvathy, J.; Gonzalez, E. A.; Asirvatham-Jeyaraj, N.; Krishna, S.; Mishra, S.; Raghu, S.; Bhati, A. S.; Tamta, A. K.; Dasgupta, S.; Kolthur-Seetharam, U.; Etchegaray, J. P.; Mostoslavsky, R.; Rao, P. S. M.; Srinivasan, N.; Sundaresan, N. R., SIRT6 transcriptionally regulates fatty acid transport by suppressing PPARgamma. Cell Rep 2021, 35 (9), 109190.
Point 7: Minor comments. Although more of a minor concern, the manuscript should be carefully reviewed for small grammatical errors (especially in the discussion).
Response 7: Thanks for your helpful suggestion. We asked Let-Pub to extensively edited the manuscript for language and grammar prior to resubmission.
We would like to thank the reviewer and hope that the revised manuscript is suitable for publication. Once again, thank you very much for your comments and suggestions.

Reviewer 2 Report
Thank you for the opportunity to review this manuscript. Overall, this is a well-written paper with an interesting result on the area in this population.
INTRODUCTION
The introduction provides sufficient background information for readers to understand the research aim, however the authors should clarify the aims of the study and the initial hypothesis.
METHODS
The methodology proposed to reach the aim of the study looks appropriate, well designed and conducted. There are few instances where assertions are made that are not substantiated with references.
RESULTS
Results paragraph include the most relevant data.
All of the tables and figures explain in a correct direction the data obtained
DISCUSSION
Discusse the different effects of different fats in diet. There is not the same fat from processed oil or fat from free animals for example.
Conclusion should respond the research aim
Explain limitation of the study and future research line according to the study conclusion
Author Response
Response to Reviewer 2 Comments
We sincerely thank the reviewer for thoroughly examining our manuscript and providing very helpful comments to guide our revision. We have tried our best to revise the manuscript according to your kind and constructive comments and suggestions. The responses to the comments are given below.
Point 1: INTRODUCTION--The introduction provides sufficient background information for readers to understand the research aim, however the authors should clarify the aims of the study and the initial hypothesis.
Response 1: Thanks for your suggestion and for your positive feedback on this study. We have indicated the purpose of the study and the initial hypothesis on lines 72–79 of the article (on page 2). Our aim was to try to understand the relationship between exercise, HFD, skd, and cardiac aging. Our initial hypothesis was that skd accelerates age-related cardiac dysfunction in Drosophila and that exercise could modulate skd expression in the heart to improve whole-body energy metabolism and cardiac function in Drosophila and prevent premature cardiac failure caused by HFD and cardiac-specific knockdown of skd.
Thanks again for your constructive suggestions.
Point 2: METHODS--The methodology proposed to reach the aim of the study looks appropriate, well designed and conducted. There are few instances where assertions are made that are not substantiated with references.
Response 2: Thanks for your positive feedback on this study.
Point 3: RESULTS-- Results paragraph include the most relevant data. All of the tables and figures explain in a correct direction the data obtained
Response 3: Thanks for your positive feedback on this study.
Point 4: DISCUSSION--Discusse the different effects of different fats in diet. There is not the same fat from processed oil or fat from free animals for example.
Response 4: Thanks for your suggestion. The high-fat protocol we used is the model that causes its widespread use in obesity. Coconut oil at 30% was added to the normal medium and HFD for 5 days. We have added the effects of different types of oils in the discussion section of the manuscript. Their different effects are discussed in terms of the composition of the oil. On pages 10–11, lines 420–426— ”Saturated fatty acids can poison normal cells. Both lard and coconut oil are rich in saturated fatty acids. Saturated fatty acids have been shown to induce apoptosis of ventricular cardiomyocytes, activation of stress-related protein kinases, and oxidative stress of proteins. The main component of olive oil is monounsaturated fatty acids. It has been shown that extra virgin olive oil diets significantly improved glycemia, insulinemia, glucose tolerance, insulin sensitivity, and insulin degradation. At a later stage, we can consider the use of different oils for the configuration of the high-fat medium.”
Thanks again for your insightful suggestions.
Point 5: Conclusion should respond the research aim
Response 5: Thanks for your suggestion. The aim of our research was to try to understand the relationship between exercise, HFD, skd, and cardiac aging. We have indicated on lines 525–529 the relationship between exercise, HFD, skd, and cardiac aging. “In conclusion, HFD impairs cardiac function and promotes metabolic disorders. Cardiac-specific knockdown of skd impairs cardiac function, disrupts lipid metabolism, and accelerates aging in Drosophila. In contrast, exercise is the remedy, and regular exercise can counteract the age-related cardiac dysfunction caused by high lipids and cardiac skd-specific knockdown.”(on page 12)
Thanks again for your constructive suggestions.
Point 6: Explain limitation of the study and future research line according to the study conclusion
Response: Thanks for your suggestion. Much work has been done in our study, but there are certainly some shortcomings. In the discussion section, we have reported them on lines 514–517 (on page 12). “However, in this experiment, a 5-day exercise intervention was performed on 12D and 36D Drosophila, with the possibility of a later exercise intervention of a different duration and different exercise lengths to further explore the mechanisms of exercise.”
In addition, it has been reported in the literature that skd in the heart regulates systemic lipid changes and is involved in the transformation of adipose tissue, suggesting that skd is involved in crosstalk in various organs. These aspects were not addressed in this experiment, and we will investigate them at a later stage. We have added new limitations to the results section of the discussion On pages 12-13, Lines 529–532. “It is worthwhile to further explore the mechanisms by which regular exercise ameliorates age-related cardiac dysfunction caused by cardiac skd-specific knockdown. It may be related to the reported involvement of skd in adipose tissue conversion. Further proof at the protein side at a later stage. ”
Thanks again for your insightful comments.
